# Prognostic impact of the combination of serum transaminase and alkaline phosphatase determined in the emergency room in patients with ST-segment elevation myocardial infarction undergoing primary percutaneous coronary intervention

**Pyung Chun Oh**[1◉], **Young Sil Eom**[2◉], **Jeonggeun Moon**[1], **Ho-Jun Jang**[3], **Tae-Hoon Kim**[3], **Jon Suh**[4], **Min Gyu Kong**[4], **Sang-Don Park**[5], **Sung Woo Kwon**[5], **Jae Yeol Choe**[6], **Soon Yong Suh**[1], **Kyounghoon Lee**[1], **Seung Hwan Han**[1], **Taehoon Ahn**[1], **Woong Chol Kang**[1]*

**1** Cardiology, Gil Medical Center, Gachon University College of Medicine, Incheon, Republic of Korea, **2** Endocrinology and Metabolism, Gil Medical Center, Gachon University College of Medicine, Incheon, Republic of Korea, **3** Cardiology, Sejong General Hospital, Bucheon, Republic of Korea, **4** Cardiology, Soon Chun Hyang University Bucheon Hospital, Bucheon, Republic of Korea, **5** Cardiology, Inha University Hospital, Incheon, Republic of Korea, **6** Department of Medicine, Gachon University School of Medicine, Incheon, Republic of Korea

◉ These authors contributed equally to this work.
* kangwch@gilhospital.com

## Abstract

### Background

Elevated serum transaminase or alkaline phosphatase (ALP) has been proposed as a novel prognosticator for ST-segment elevation myocardial infarction (STEMI). We evaluated the combined prognostic impact of elevated serum transaminases and ALP on admission in STEMI patients who underwent primary percutaneous coronary intervention (PCI).

### Methods

A total of 1176 patients with STEMI undergoing primary PCI were retrospectively enrolled from the INTERSTELLAR registry. Hypoxic liver injury (HLI) was defined as serum transaminase > twice the upper limit of normal. The cut-off value of high ALP was set at the median level (73 IU/L). Patients were divided into four groups according to their serum transaminase and ALP levels. The primary endpoint was major adverse cardiac or cerebrovascular events (MACCE), defined as the composite of all-cause death, non-fatal myocardial infarction, non-fatal stroke, and ischemia-driven revascularization.

### Results

Median follow-up duration was 25 months (interquartile range, 10–39 months). The rate of MACCE was highest in patients with HLI (+) and high ALP (25.9%), compared to those in the

**Data Availability Statement:** All relevant data are within the manuscript and its Supporting Information files.

**Funding:** This work was supported by the Yuhan Corporation, Republic of Korea (P.C.O.). The funder had no role in study design, data collection and analysis, decision to publish, or preparation of the manuscript. No additional external funding was received for this study.

**Competing interests:** The authors have declared that no competing interests exist.

other groups (8.2% in HLI [-] and low ALP, 11.8% in HLI [-] and high ALP, and 15.0% in HLI [+] and low ALP). Each of HLI or high ALP was an independent predictor for MACCE (HR 1.807, 95% CI 1.191–2.741; HR 1.721, 95% CI 1.179–2.512, respectively). Combined HLI and high ALP was associated with the worst prognosis (HR 3.145, 95% CI 1.794–5.514).

## Conclusions

Combined HLI and high ALP on admission is associated with poor clinical outcomes in patients with STEMI who have undergone primary PCI.

## Introduction

Although primary PCI performed in a timely fashion for ST-segment elevation myocardial infarction (STEMI) has improved clinical outcomes, the mortality rate of patients with STEMI is still high [1]. Therefore, it is important to identify patients at a high risk for major adverse cardiovascular events after STEMI using early risk stratification at the time of presentation.

Liver function tests including serum transaminases and alkaline phosphatase (ALP) are usually measured in the emergency room as part of routine blood tests. In patients with acute myocardial infarction (MI), abnormal results of liver function parameters are often observed without clinically identifiable liver disease [2]. Recent studies suggest that nonalcoholic fatty liver disease fibrosis score including serum transaminases is a significant predictor of mortality in general population [3, 4]. This score is associated with increased risk of recurrent cardiovascular events in post-acute coronary syndrome patients [5]. In addition, there were a few reports that increased ALP is associated with poor clinical outcomes in patients with STEMI [6, 7]. We also reported that elevated ALP level at presentation, but within the upper limit of normal, was found to be independently associated with higher risk of major adverse events after primary PCI for STEMI [8]. In addition, our group revealed that mild elevation of serum transaminases, resulting from hypoxic liver injury (HLI), may predict early mortality in STEMI treated with primary PCI [9]. HLI with contrast-induced nephropathy was also associated with worse outcome in STEMI [10]. Measurement of these parameters may be an early and easily available investigation in most cases of STEMI before the evaluation of cardiac function and coronary anatomy. Thus, we evaluated the prognostic impact of a combination of liver function parameters including serum transaminases and ALP at the time of presentation in patients with STEMI undergoing primary PCI.

## Methods

### Study population

This is a retrospective multi-center study using data obtained from the "INcheon-Bucheon cohorT of patients undERgoing primary PCI for acute ST-ELevation myocardiaL infARction (INTERSTELLAR)" registry. The INTERSTELLAR registry is a retrospective, observational, 4-regional hospital based registry that reflects current management practices, risk factors, and clinical outcomes in patients with STEMI undergoing primary PCI in the Incheon-Bucheon province (clinicaltrials.gov identifier NCT02804958) [8, 11]. A total of 1537 consecutive patients with STEMI who underwent primary PCI between 2007 and 2014 were enrolled. The Institutional Review Boards (Gachon University Gil Medical Center Institutional Review Board, Sejong General Hospital Institutional Review Board, Soon Chun Hyang University

Bucheon Hospital Institutional Review Board, and Inha University Hospital Institutional Review Board) of the four participating hospitals approved the study protocol. Patients previously diagnosed with coronary artery disease, cardiomyopathy, valvulopathy ($\geq$ moderate), pericardial disease, or congenital heart disease were excluded (n = 71), as were patients with established liver disease, bone disease, or chronic kidney disease capable of affecting levels of serum transaminases or ALP (n = 275), and patients who had not undergone initial liver function tests (n = 13). Chronic kidney disease was defined as an estimated glomerular filtration rate of < 60 mL/min/1.73 m$^2$, as determined using the 4-variable Modification of Diet in Renal Disease formula. After applying the above-mentioned inclusion and exclusion criteria, 1178 (82.7% male, mean age 58.2 ± 12.3 years) of the originally selected 1537 patients were included in the present study. The baseline risk factors, laboratory parameters, echocardiographic and angiographic findings, length of follow-up, and details of adverse events were recorded.

## Primary PCI and in-hospital management

All patients visited the emergency room first without being admitted directly to the catheterization laboratory. All procedures were performed according to current standard guidelines. Before PCI, patients were pre-medicated with aspirin (at least 100 mg), and a loading dose of a P2Y12 receptor antagonist was administered. Heparin was administered throughout the procedure in order to maintain an activated clotting time of 250 seconds or longer. A glycoprotein IIb/IIIa receptor blocker was administered at the discretion of the operator. Coronary angiography was performed using standard techniques. Decisions on whether to use thrombectomy devices, intravascular ultrasound, an intra-aortic balloon pump, and percutaneous cardiopulmonary support were made by the operator. Procedural success was defined as no in-hospital death, no emergency bypass surgery, and the achievement of Thrombolysis in Myocardial Infarction (TIMI) flow grade 3 in the distal portion of the infarct-related artery and the presence of < 30% residual stenosis. After primary PCI, all patients were monitored in a coronary care unit for at least 24 hours. Two-dimensional transthoracic echocardiography was performed within 12 hours of the index procedure. Standard medical management was provided by responsible physicians.

## Follow-up data acquisition

The primary study outcome was a major adverse cardiac or cerebrovascular event (MACCE), defined as the composite of all-cause death, non-fatal MI, non-fatal stroke, and ischemia-driven revascularization. Patient follow-up data were collected using a review of all medical records and/or standardized telephone interviews.

## Patient classification

Patients were divided into four groups according to their serum transaminase and ALP levels determined in the emergency room: HLI (-) and low ALP, HLI (-) and high ALP, HLI (+) and low ALP, and HLI (+) and high ALP. HLI was defined as a serum transaminases level > twice the upper limit of normal (i.e., aspartate transaminase [AST] > 80 U/L or alanine transaminase [ALT] > 80 U/L), as per Moon et al [9]. The cut-off point of high ALP group was set at the median level (73 IU/L) in this study.

## Statistical analysis

Continuous variables are presented as mean ± standard deviation for normally distributed data or as median (interquartile range) for skewed data. Categorical variables are outlined

using absolute and relative (percentage) frequencies. The four groups were compared with respect to baseline characteristics and adverse clinical outcomes using one-way analysis of variance for continuous variables or Pearson's $\chi^2$ test for categorical variables. Event-free survival rates were estimated using the Kaplan-Meier product-limit estimation method with the log-rank test. Multivariate Cox regression analysis was performed to quantify relationships between covariates previously reported to be related with adverse outcomes or liver function parameters and time to events. We used 2 models for multivariate analysis. In model 1, conventional risk factors for adverse events, such as age, male sex, diabetes mellitus, hypertension, reduced ejection fraction (< 40%), Killip class, anterior MI, symptom to balloon time and peak creatine kinase-myocardial band isoenzyme (CK-MB) were used as covariates. In model 2, the following extra covariates were added to those in model 1: multi-vessel disease, estimated glomerular filtration rate, albumin, total bilirubin, glucose, calcium, and phosphate. P values of less than 0.05 were considered statistically significant, and the analysis was performed using SPSS version 20 (SPSS, Chicago, IL, USA) and SAS University Edition (SAS Institute, Cary, NC, USA).

## Results

### Baseline characteristics

The demographic and laboratory data measured at the time of presentation are shown in Table 1. Of the 1176 patients, 473 patients (40.2%) were in the HLI (-) & low ALP group, 474 patients (40.3%) were in the HLI (-) & high ALP group, 113 patients (9.6%) were in the HLI (+) & low ALP group, and 116 patients (9.9%) were in the HLI (+) & high ALP group. The prevalence of HLI in patients with STEMI was 19.5%. Elevation of serum ALT or AST > twice the upper limit of normal was in 71 patients (6.0%) or 205 patients (17.4%), respectively. ALP levels ranged from 14 to 192 IU/L (median 73 IU/L, interquartile range 61–79 IU/L). Patients with low ALP level were more likely to be male and patients with HLI tended to be Killip class > II. Moreover, patients with HLI and high ALP level were more likely to have higher glucose level at the time of presentation. However, age, presence of hypertension or diabetes mellitus, smoking status, blood pressure, or prevalence of anterior MI did not differ significantly among the 4 groups. There were no significant differences of discharge medications including beta-blocker, renin-angiogensin system blocker and statin among the 4 groups (S1 Table).

A summary of angiographic, procedural, and echocardiographic data is depicted in Table 2. The symptom to balloon time was significantly longer in patients with HLI than in patients without HLI. Patients with HLI and high ALP level tended to have a lower rate of procedural success. Left ventricular ejection fraction was significantly lower in patients with HLI compared to that in patients without HLI.

### Clinical outcomes according to the presence of HLI and ALP level

The incidence of adverse events is summarized in Table 3. The overall rate of in-hospital mortality was 3.1% in patients with STEMI who underwent primary PCI. The median duration of hospitalization for patients with in-hospital death was 4.0 days (interquartile range, 1.0–16.5 days). The in-hospital mortality in patients with HLI and high ALP level (9.5%) was significantly higher than that in each of the groups with HLI (-) and low ALP (1.7%), HLI (-) and high ALP (2.3%), and HLI (+) and low ALP (6.2%).

Median duration of follow-up was 25 months (interquartile range, 10–39 months). MACCE occurred in 12.1% of patients. The rate of MACCE was the highest in patients with HLI and high ALP (25.9%), compared to those in the other groups (8.20%, 11.8%, and 15.0%

**Table 1. Demographic and laboratory data.**

| | All (n = 1176) | HLI[†] (-) & low ALP (n = 473) | HLI (-) & high ALP (n = 474) | HLI (+) & low ALP (n = 113) | HLI (+) & high ALP (n = 116) | p value |
|---|---|---|---|---|---|---|
| *Demographic data* | | | | | | |
| Age, years | 58.3 ± 12.4 | 57.8 ± 12.5 | 58.8 ± 11.8 | 56.8 ± 12.9 | 59.1 ± 13.3 | 0.401 |
| Men, n (%) | 973 (82.7) | 406 (85.8) | 377 (79.5) | 99 (87.6) | 91 (78.4) | 0.019 |
| Body mass index, kg/m$^2$ | 24.2 ± 3.2 | 24.3 ± 3.1 | 24.0 ± 3.4 | 24.4 ± 2.9 | 23.8 ± 3.6 | 0.276 |
| Current smoker, n (%) | 677 (57.6) | 268 (56.7) | 277 (58.4) | 61 (54.0) | 71 (61.7) | 0.791 |
| Diabetes mellitus, n (%) | 265 (22.5) | 106 (22.4) | 112 (23.6) | 25 (22.1) | 22 (19.0) | 0.756 |
| Hypertension, n (%) | 507 (43.1) | 204 (43.1) | 213 (44.9) | 41 (36.3) | 49 (42.2) | 0.419 |
| Systolic blood pressure, mm Hg | 126.8 ± 27.4 | 125.4 ± 25.2 | 128.5 ± 28.8 | 123.7 ± 30.0 | 128.3 ± 27.3 | 0.164 |
| Diastolic blood pressure, mm Hg | 77.8 ± 17.5 | 76.2 ± 16.2 | 78.9 ± 18.2 | 76.5 ± 18.3 | 80.8 ± 17.9 | 0.501 |
| Heart rate, per minute | 77.1 ± 18.8 | 74.9 ± 18.6 | 77.1 ± 18.2 | 80.6 ± 20.1 | 83.0 ± 19.9 | 0.183 |
| Killip class >II, n (%) | 106 (9.0) | 36 (7.6) | 39 (8.2) | 14 (12.4) | 17 (14.9) | 0.050 |
| Anterior wall MI, n (%) | 614 (52.7) | 233 (49.6) | 256 (54.7) | 61 (54.5) | 64 (55.2) | 0.389 |
| *Laboratory data* | | | | | | |
| Albumin, d/dL | 4.2 ± 0.4 | 4.2 ± 0.4 | 4.3 ± 0.4 | 4.2 ± 0.4 | 4.2 ± 0.4 | 0.448 |
| Glucose, mg/dL | 170.1 ± 75.6 | 162.8 ± 63.3 | 175.1 ± 74.1 | 169.1 ± 85.2 | 180.2 ± 108.6 | <0.001 |
| Total bilirubin, mg/dL | 0.6 ± 0.3 | 0.6 ± 0.3 | 0.6 ± 0.3 | 0.8 ± 0.3 | 0.7 ± 0.3 | <0.001 |
| AST, IU/L | 32.0 (23.0–61.0) | 27.0 (21.0–37.0) | 28.0 (22.0–37.0) | 166.0 (97.5–262.0) | 133.0 (93.0–199.0) | <0.001 |
| ALT, IU/L | 27.0 (19.0–41.0) | 23.0 (17.0–33.0) | 23.0 (18.0–34.0) | 59.0 (36.0–84.0) | 58.0 (38.0–87.0) | <0.001 |
| ALP, IU/L | 73.0 (61.0–91.0) | 60.0 (54.0–67.0) | 88.9 (80.0–104.4) | 62.0 (53.5–67.0) | 93.0 (79.0–110.3) | <0.001 |
| Creatinine, mg/dL | 0.94 ± 0.18 | 0.97 ± 0.17 | 0.93 ± 0.19 | 0.93 ± 0.19 | 0.94 ± 0.18 | 0.474 |
| Estimated GFR, mL/min/1.73 m$^2$ | 87.2 ± 24.5 | 85.0 ± 17.7 | 88.5 ± 24.8 | 91.2 ± 22.5 | 86.5 ± 19.0 | 0.035 |
| Calcium, mg/dL | 8.9 ± 0.5 | 8.9 ± 0.5 | 9.0 ± 0.5 | 8.8 ± 0.6 | 8.9 ± 0.7 | 0.261 |
| Phosphate, mg/dL | 3.2 ± 1.5 | 3.1 ± 0.8 | 3.1 ± 2.1 | 3.5 ± 1.0 | 3.4 ± 1.0 | 0.776 |
| Initial CK-MB, ng/mL | 4.9 (2.0–23.1) | 3.3 (1.6–8.8) | 4.0 (1.8–12.4) | 94.7 (34.3–155.7) | 68.7 (8.2–192.5) | <0.001 |
| Peak CK-MB, ng/mL | 175.5 (78.2–300.0) | 160.6 (69.4–300.0) | 176.4 (74.5–300.0) | 179.3 (95.8–300.0) | 232.0 (106.3–300.0) | <0.001 |

HLI, hypoxic liver injury; ALP, alkaline phosphatase; AST, aspartate transaminase; ALT, alanine transaminase; GFR, glomerular filtration rate; CK-MB, creatine kinase-myocardial band isoenzyme; hs-CRP, high-sensitivity C-reactive protein.

[†]Hypoxic liver injury (HLI) was defined as an elevation of serum transaminase levels to more than twice the upper limit of normal.

in the groups of HLI [-] and low ALP, HLI [-] and high ALP level, and HLI [+] and low ALP, respectively; p < 0.001). When the presence of HLI was defined based on serum ALT only, there was a significant trend to higher rate of in-hospital mortality or MACCE in patients with HLI or high ALP and the rate of MACCE was the highest in patients with HLI and high ALP (S2 Table).

Kaplan-Meier survival curves showed that the patients with HLI and high ALP level experienced significantly more MACCE than the patients of HLI (-) and low ALP level as determined by the log-rank test (Fig 1). Among patients without HLI, MACCE-free survival rates within a 14-month follow-up period in patients with high ALP and those with low ALP were similar. However, thereafter there was a significant divergence of MACCE-free survival rate between the 2 groups. Regardless of ALP level, patients with HLI had poor clinical outcomes soon after diagnosis of STEMI, compared to those without HLI. Among the composite of adverse events, there was a significant difference in all-cause mortality rate between the 4 groups (p < 0.001).

**Table 2. Angiographic, procedural and echocardiographic data.**

| | All (n = 1176) | HLI (-) & low ALP (n = 473) | HLI (-) & high ALP (n = 474) | HLI (+) & low ALP (n = 113) | HLI (+) & high ALP (n = 116) | p value |
|---|---|---|---|---|---|---|
| *Angiographic and procedural data* | | | | | | |
| Infarct related artery, n (%) | | | | | | |
| Left main | 9 (0.8) | 2 (0.4) | 5 (1.1) | 2 (1.8) | 0 (0.0) | 0.532 |
| Left anterior descending | 605 (51.9) | 231 (49.1) | 251 (53.6) | 59 (52.7) | 64 (55.2) | |
| Left circumflex | 131 (11.2) | 58 (12.3) | 50 (10.7) | 14 (12.5) | 9 (7.8) | |
| Right coronary | 421 (36.1) | 179 (38.1) | 162 (34.6) | 37 (33.0) | 43 (37.1) | |
| Extent of coronary artery disease, n (%) | | | | | | |
| 1-vessel | 496 (42.5) | 205 (43.7) | 195 (41.6) | 48 (42.9) | 48 (41.4) | 0.390 |
| 2-vessel | 386 (33.1) | 157 (33.5) | 144 (30.7) | 41 (36.6) | 44 (37.9) | |
| 3-vessel | 284 (24.4) | 107 (22.8) | 130 (27.7) | 23 (20.5) | 24 (20.7) | |
| Baseline TIMI flow grade, n (%) | | | | | | |
| 0–2 | 1067 (91.2) | 429 (91.1) | 427 (90.7) | 102 (91.1) | 109 (94.0) | 0.732 |
| 3 | 103 (8.8) | 42 (8.9) | 44 (9.3) | 10 (8.9) | 7 (6.0) | |
| Final TIMI flow grade, n (%) | | | | | | |
| 0–2 | 121 (10.4) | 49 (10.4) | 42 (8.9) | 10 (8.9) | 20 (17.2) | 0.065 |
| 3 | 1048 (89.6) | 421 (89.6) | 429 (91.1) | 102 (91.1) | 96 (82.8) | |
| Door-to-balloon time, minute | 71.0 (58.0–87.0) | 70.0 (57.0–85.0) | 69.0 (57.0–86.0) | 81.0 (64.5–111.0) | 72.0 (59.8–86.3) | 0.002 |
| Symptom-to-balloon time, minute | 206.0 (132.5–397.0) | 177.0 (125.0–315.0) | 198.0 (127.0–330.0) | 440.0 (200.5–1114.0) | 363.5 (160.5–940.8) | <0.001 |
| Procedural success, n (%) | 1048 (89.6) | 421 (89.6) | 429 (91.1) | 102 (91.1) | 96 (82.8) | 0.065 |
| *Echocardiographic data* | | | | | | |
| LVEF, % | 49.3 ± 11.5 | 50.4 ± 11.2 | 50.0 ± 11.4 | 46.8 ± 11.2 | 44.9 ± 12.2 | <0.001 |
| LVEDD, mm | 51.1 ± 5.4 | 51.4 ± 5.7 | 50.5 ± 5.0 | 53.1 ± 5.0 | 51.1 ± 5.4 | 0.014 |
| LAVI, mL/m$^2$ | 19.0 ± 8.2 | 19.5 ± 9.0 | 18.7 ± 7.7 | 17.1 ± 7.9 | 19.3 ± 7.1 | 0.551 |
| E/E' | 11.6 ± 4.9 | 11.3 ± 4.7 | 11.6 ± 5.0 | 11.9 ± 4.9 | 12.2 ± 5.0 | 0.685 |

HLI, hypoxic liver injury; ALP, alkaline phosphatase; LVEF, left ventricular ejection fraction; LVEDD, left ventricular end-diastolic dimension; LAVI, left atrial volume index.

**Table 3. Incidence of adverse clinical outcomes.**

| | All (n = 1176) | HLI (-) & low ALP (n = 473) | HLI (-) & high ALP (n = 474) | HLI (+) & low ALP (n = 113) | HLI (+) & high ALP (n = 116) | p value |
|---|---|---|---|---|---|---|
| In-hospital death | 37 (3.1) | 8 (1.7) | 11 (2.3) | 7 (6.2) | 11 (9.5) | <0.001 |
| MACCE | 142 (12.1) | 39 (8.2) | 56 (11.8) | 17 (15.0) | 30 (25.9) | <0.001 |
| All-cause death | 64 (5.4) | 16 (3.4) | 19 (4.0) | 11 (9.7) | 18 (15.5) | <0.001 |
| Non-fatal myocardial infarction | 36 (3.1) | 8 (1.7) | 22 (4.6) | 3 (2.7) | 3 (2.6) | 0.068 |
| Ischemia-driven revascularization | 30 (2.6) | 10 (2.1) | 13 (2.7) | 2 (1.8) | 5 (4.3) | 0.541 |
| Non-fatal stroke | 12 (1.0) | 5 (1.1) | 2 (0.4) | 1 (0.9) | 4 (3.4) | 0.037 |

HLI, hypoxic liver injury; ALP, alkaline phosphatase; MACCE, major adverse cardiac and cerebrovascular events.

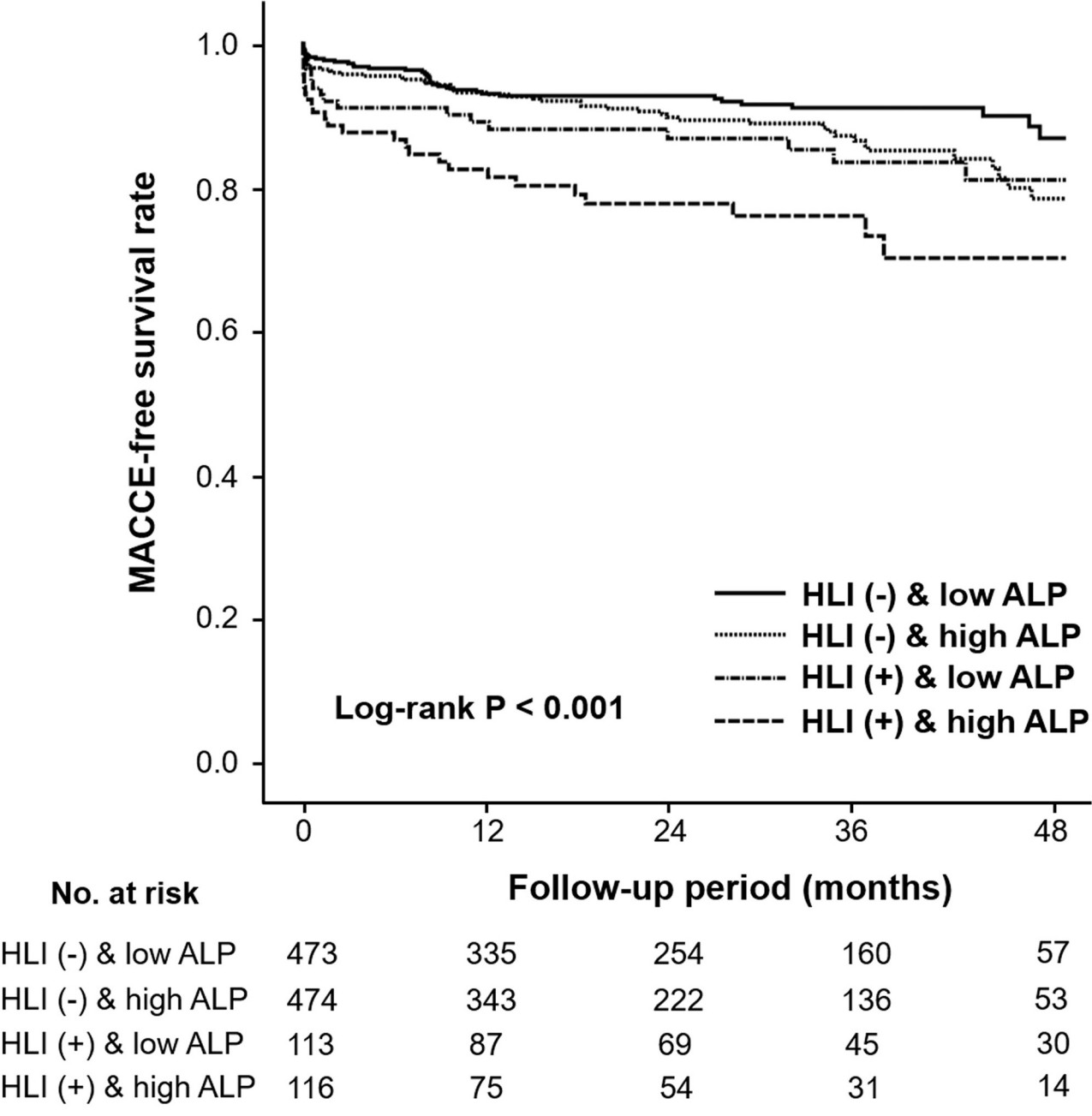

**Fig 1. Major adverse cardiac or cerebrovascular event (MACCE)-free survival curve according to risk groups defined by combination of hypoxic liver injury and serum alkaline phosphatase (ALP) level.**

### Predictors for MACCE

In the multivariate Cox proportional hazards analysis for MACCE, age (HR 1.226 per 10-year increase, 95% CI 1.040–1.445, p = 0.015), Killip class > II (HR 1.860, 95% CI 1.081–3.200, p = 0.025), LVEF < 40% (HR 1.684, 95% CI 1.100–2.576, p = 0.016), presence of HLI (HR 1.807, 95% CI 1.191–2.741, p = 0.005), and high ALP > 73 IU/L (HR 1.721, 95% CI 1.179–2.512, p = 0.005) were significant risk factors (Table 4). Furthermore, the combination of HLI

**Table 4. Predictors of major adverse cardiac and cerebrovascular events.**

| Variables | Univariate analysis | | | Multivariate analysis | | |
|---|---|---|---|---|---|---|
| | HR | 95% CI | p value | HR | 95% CI | p value |
| Age (per 10-year increase) | 1.252 | 1.094–1.434 | 0.001 | 1.226 | 1.040–1.445 | 0.015 |
| Male | 1.171 | 0.742–1.847 | 0.497 | | | |
| Diabetes mellitus | 1.289 | 0.879–1.891 | 0.194 | | | |
| Hypertension | 1.089 | 0.779–1.521 | 0.618 | | | |
| Killip class > II | 2.880 | 1.891–4.388 | <0.001 | 1.860 | 1.081–3.200 | 0.025 |
| Anterior infarction | 1.238 | 0.885–1.731 | 0.212 | | | |
| LVEF < 40% | 2.559 | 1.759–3.723 | <0.001 | 1.684 | 1.100–2.576 | 0.016 |
| Symptom to balloon time (log 10) | 1.138 | 0.774–1.673 | 0.512 | | | |
| Peak CK-MB (log 10) | 1.453 | 1.017–2.075 | 0.040 | | | |
| Hypoxic liver injury | 1.965 | 1.381–2.796 | <0.001 | 1.807 | 1.191–2.741 | 0.005 |
| High ALP > 73 IU/L | 1.718 | 1.224–2.412 | 0.002 | 1.721 | 1.179–2.512 | 0.005 |

HR, hazard ratio; CI, confidence interval; LVEF, left ventricular ejection fraction; CK-MB, creatine kinase-myocardial band isoenzyme; ALP, alkaline phosphatase

and high ALP was associated with the worst prognosis, even after adjusting several covariates in the 2 models (HR 3.145 using model 1, 95% CI 1.794–5.514, p < 0.001), compared to the reference group of HLI (-) and low ALP (Table 5, Fig 2). C-statistics for predicting MACCE was 0.656 (95% CI, 0.604–0.708) in the multivariate Model 1 excepting HLI status and high ALP. The addition of HLI status and high ALP to the Model 1 significantly improved the c-statistics to 0.685 (95% CI, 0.635–0.736; p for difference = 0.044) (S1 Fig).

## Discussion

The main findings of this study were as follows: (1) HLI (defined as a level of serum transaminases > twice the upper limit of normal) diagnosed in the emergency room was common (19.4%) and most patients (about 95%) had a serum ALP level below the upper limit of normal in STEMI. (2) Each of HLI or high ALP (> median level, 73 IU/L) was found to be independently associated with the risk of MACCE after primary PCI in STEMI patients. (3) The patients with HLI and high ALP had the worst clinical outcomes including in-hospital

**Table 5. Combined predictive value of hypoxic liver injury (HLI) and alkaline phosphatase (ALP) level for major adverse cardiac and cerebrovascular events (MACCE).**

| | Model 1[†] | | | Model 2[‡] | | |
|---|---|---|---|---|---|---|
| | HR | 95% CI | p value | HR | 95% CI | p value |
| HLI (-) & low ALP | Reference | | | Reference | | |
| HLI (-) & high ALP | 1.631 | 1.037–2.565 | 0.034 | 1.582 | 0.996–2.513 | 0.052 |
| HLI (+) & low ALP | 1.625 | 0.843–3.130 | 0.147 | 1.321 | 0.645–2.707 | 0.447 |
| HLI (+) & high ALP | 3.145 | 1.794–5.514 | <0.001 | 2.712 | 1.497–4.913 | 0.001 |

HR, hazard ratio; CI, confidence interval

[†]Model 1: HRs have been adjusted for age, sex, diabetes mellitus, hypertension, ejection fraction (< 40%), Killip class (> II), anterior myocardial infarction, symptom to balloon time (log 10) and peak creatine kinase-myocardial band isoenzyme (log 10).

[‡]Model 2: HRs have been adjusted for Model 1 variables and additional covariates as follows: multi-vessel disease, estimated glomerular filtration rate, albumin, total bilirubin, and glucose.

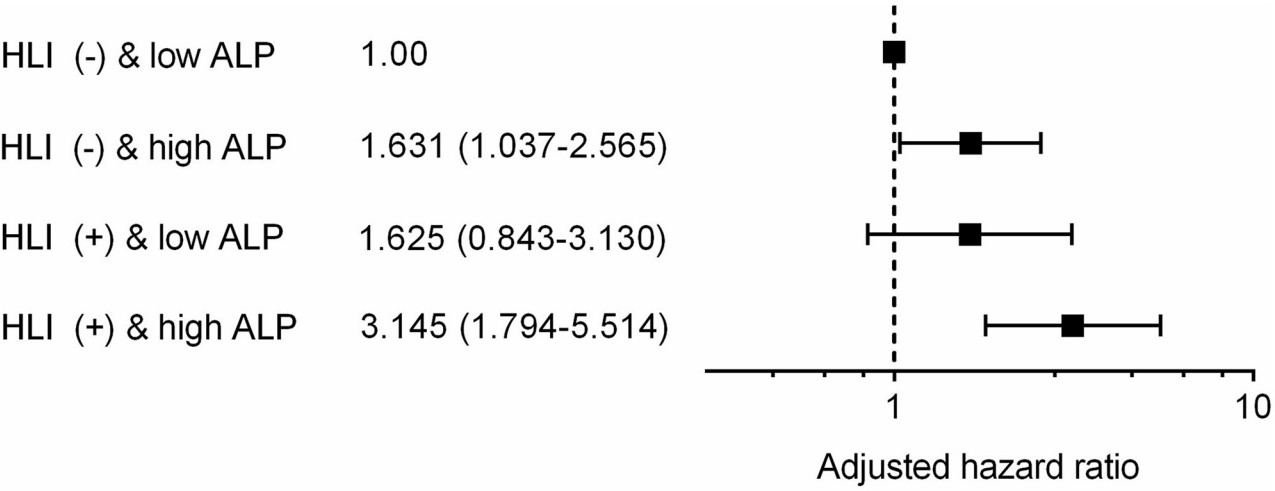

| | Adjusted hazard ratio |
|---|---|
| HLI (-) & low ALP | 1.00 |
| HLI (-) & high ALP | 1.631 (1.037-2.565) |
| HLI (+) & low ALP | 1.625 (0.843-3.130) |
| HLI (+) & high ALP | 3.145 (1.794-5.514) |

**Fig 2. Predictive value of risk groups defined by combination of hypoxic liver injury (HLI) and serum alkaline phosphatase (ALP) level.**

mortality and MACCE, suggesting a synergistic effect of HLI and serum ALP level as early prognosticators in STEMI.

Our group previously reported that each of HLI or high ALP level on admission was independently related with poor clinical outcomes in STEMI patients from a single-center cohort or the same cohort of this study, respectively [8, 9]. In addition, combined HLI and dysglycemia (hyper or hypoglycemia) had a synergistic impact to predict in-hospital mortality in STEMI [12]. In a study of subgroup with follow-up serum creatinine data (n = 668) in this cohort, HLI with contrast-induced nephropathy was also associated with the worst outcome in STEMI [10]. In the current study, we evaluated the combined predictive value of liver function parameters including serum transaminases and ALP for MACCE. To our knowledge, this is the first study to report the usefulness of combined serum transaminases and ALP determined in the emergency room as predictors for MACCE in STEMI.

## Prognostic implication of HLI

The liver is an active vital organ that is very sensitive to hemodynamic changes due to its complex vascular system and high metabolic activity [13]. In fact, abnormal results of measurement of serum transaminases are often observed in patients with acute MI [2]. Serum AST has been traditionally used as a biomarker for myocardial injury before the advent of biomarkers more specific to the heart such as CK-MB and troponin [14]. Although the source of elevated serum transaminases, especially AST, might be ischemic myocardial tissue *per se*, the rise in the levels of these enzymes may often result from hypoxic liver damage owing to both impaired forward perfusion and passive backward congestion, which are prevalent during STEMI [15]. Recently, there have been a few reports stating that the elevation of serum transaminases was independently associated with poor clinical outcomes in patients with STEMI. We revealed that elevation of serum transaminases to higher than twice the upper limit of normal, denominated as HLI, was associated with higher rates of mortality and major adverse cardiovascular events in STEMI patients after primary PCI based on cohort data from a single center [9]. Of note, HLI was strongly correlated with post-PCI left ventricular systolic function and was superior to CK-MB in predicting outcomes in STEMI. In the current study, we reaffirmed that HLI had prognostic value as an early predictor of adverse events using 4-hospital cohorts of STEMI

patients, whereas peak CK-MB did not. The failure of peak CK-MB to predict poor outcomes may be attributed to survival bias. The correlation between HLI and poor outcomes was significant even after adjusting for several variables reflecting severity of MI. Several factors may contribute to the poor prognostic value of HLI on admission. First, levels of serum transaminases might reflect the severity and clinical significance of MI. Both serum transaminases correlated well with the area under the curve of CK-MB and were independent predictors of all-cause mortality and adverse events after adjustment for CK-MB in patients with STEMI [16]. In the study using cardiac magnetic resonance, admission and peak levels of serum transaminases were significantly associated with left ventricular ejection fraction, infarct size, and the presence of microvascular damage after primary PCI in STEMI patients [6]. Second, HLI on admission may signify a delay from symptom onset to hospital presentation after occlusion of the infarct-related coronary artery. It may imply extensive and irreversible myocardial damage. Finally, HLI may be a very sensitive marker for possibly transient but significant left ventricular dysfunction and reduced cardiac output before primary PCI in STEMI.

## Prognostic implication of serum ALP level

ALP is actually enzyme found primarily liver and bone, and to lesser extents in intestine, placenta, and kidneys. Accordingly, serum ALP is used as a marker of hepatic or bony disease in clinical practice. However, serum ALP level was not significantly elevated in most patients with coronary artery disease requiring PCI [17]. In the present study, most patients were also within the normal ALP range, suggesting that ischemic hepatopathy might play a more dominant role than congestive hepatopathy in STEMI with low cardiac output. It has been suggested that ALP plays a pivotal role in mineral metabolism and might be a biochemical marker of vascular calcification [17–19]. Recent reports have shown that a significant association exists between elevated ALP levels and cardiovascular events and mortality in various populations, such as, hemodialysis patients, survivors of stroke or MI, and the elderly [7, 17, 19, 20]. We also revealed that STEMI patients in the highest ALP tertile were independently associated with a higher risk of MACCE after primary PCI [8]. Although the underlying mechanisms for association between elevated ALP levels and poor clinical outcomes in patients with acute MI are unclear, a putative link between ALP and vascular calcification might be a possible explanation. ALP was upregulated in peripheral arteries with medial calcification [21]. In addition, there was an independent association between higher ALP level and coronary artery calcification in hemodialysis patients [22]. In turn, adverse clinical outcomes might be mediated by the deleterious effects of vascular calcification on plaque stability, vascular stiffness, valvular heart disease, and calciphylaxis [18]. However, bone-specific ALP levels were not available in this study, and this mechanism remains speculative.

## Prognostic implication of combined HLI and high ALP level

The combined predictive value of HLI and high ALP for adverse events in STEMI patients who have undergone primary PCI has not been reported before. In the present study, patients with HLI and high ALP on admission had significantly higher risk of all-cause mortality and MACCE during a 25-month follow-up period independently of MI severity, compared to patients with no HLI and low ALP. In addition, the combination of HLI and high ALP showed a higher MACCE rate than either one alone, suggesting a synergistic effect of the 2 factors. Among patients without HLI, there was no significant difference in MACCE rate within 1 year after the index event between the high ALP and low ALP groups. This suggests that ALP level might have predictive value in only later stages (> 1 year) after MI in patients without HLI.

However, high ALP level was significantly related to poor outcomes in both the early and late stages of MI among patients with HLI.

Both serum transaminases and ALP are usually measured on admission as part of routine laboratory testing. These are simple, easily available, and inexpensive. The presence/absence of HLI and the serum ALP level can be determined even before primary PCI and echocardiography in the emergency room. Even small increases in serum transaminases might represent possibly transient but significant LV dysfunction. In addition, serum ALP level might be viewed as a potential marker of the burden of vascular disease related to arterial calcification. Therefore, the combined information on HLI and serum ALP level, albeit within the reference range, predicts not only in-hospital mortality but also the probability of long-term future events. To date, no direct treatment for HLI or high ALP is available. High risk patients with HLI and high ALP level are most likely to derive benefit from early aggressive treatment for cardiac pump failure and cardiovascular risk factors, such as early initiation of renin-angiotension system blocker or high-intensity statin. However, a potential method to improve the prognosis of patients based on the HLI status and serum ALP levels is beyond the scope of this study.

## Limitations

Our study has several limitations. First, this is a retrospective and observational study that consisted of a Korean population, and extrapolation to other parts of the world may not be valid. Second, we cannot preclude the possibility of residual confounding factors, although multivariate analysis was performed for adjusting for multiple potential confounders. Third, whether serum transaminases, especially AST, were elevated solely due to hypoxic liver damage is not certain. Serum ALT is a more specific marker of liver injury. It might be more reasonable that the definition of HLI have to be based on serum ALT rather than AST. In this study, the incidence of HLI based on serum ALT only was much less than that based on both serum ALT and AST. Therefore, larger study will be needed to evaluate the prognostic impact of serum ALT level. In addition, the cut-off value of HLI was not fully validated. Fourth, serum transaminases and ALP were not measured using a central laboratory. Fifth, we did not obtain serial data of serum transaminases and ALP levels during the hospitalization. Peak values of serum transaminases were significantly more related with in-hospital mortality than admission values in STEMI patients [23]. It was impossible to determine its prognostic implication. Finally, we had to divide the study population into low and high ALP groups using the median value of serum ALP, which may have introduced statistical inaccuracy.

## Supporting information

**S1 Table. Discharge medication data among MI survivors.**
(PDF)

**S2 Table. Incidence of adverse clinical outcomes according to levels of serum ALT and ALP.**
(PDF)

**S1 Fig. Receiver operating curves for the predicted probabilities of selected risk models before (blue line) and after (red line) the addition of hypoxic liver injury (HLI) and alkaline phosphatase (>73 IU/L) status to the multivariate Model 1.** [†]The Model 1 included age, sex, diabetes mellitus, hypertension, ejection fraction, Killip class, anterior myocardial infarction, symptom to balloon time (log 10) and peak creatine kinase-myocardial band isoenzyme (log 10).
(PDF)

## Author Contributions

**Conceptualization:** Pyung Chun Oh, Kyounghoon Lee, Woong Chol Kang.

**Data curation:** Pyung Chun Oh, Jeonggeun Moon, Ho-Jun Jang, Tae-Hoon Kim, Jon Suh, Min Gyu Kong, Sang-Don Park, Sung Woo Kwon, Jae Yeol Choe, Soon Yong Suh, Kyounghoon Lee, Seung Hwan Han, Taehoon Ahn, Woong Chol Kang.

**Formal analysis:** Pyung Chun Oh, Kyounghoon Lee, Woong Chol Kang.

**Funding acquisition:** Woong Chol Kang.

**Investigation:** Pyung Chun Oh, Jeonggeun Moon, Ho-Jun Jang, Tae-Hoon Kim, Jon Suh, Min Gyu Kong, Sang-Don Park, Sung Woo Kwon, Kyounghoon Lee, Woong Chol Kang.

**Methodology:** Pyung Chun Oh, Ho-Jun Jang, Tae-Hoon Kim, Jon Suh, Kyounghoon Lee, Woong Chol Kang.

**Project administration:** Woong Chol Kang.

**Resources:** Woong Chol Kang.

**Software:** Woong Chol Kang.

**Supervision:** Kyounghoon Lee, Woong Chol Kang.

**Validation:** Kyounghoon Lee, Woong Chol Kang.

**Visualization:** Woong Chol Kang.

**Writing – original draft:** Pyung Chun Oh, Young Sil Eom.

**Writing – review & editing:** Kyounghoon Lee, Woong Chol Kang.

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
