## [Decision Letter · Decision Letter 0]

16 Apr 2020

PONE-D-19-32757

Prognostic impact of the combination of serum transaminase and alkaline phosphatase determined in the emergency room in patients with ST-segment elevation myocardial infarction undergoing primary percutaneous coronary intervention

PLOS ONE

Dear MD Kang,

Thank you for submitting your manuscript to PLOS ONE. After careful consideration, we feel that it has merit but does not fully meet PLOS ONE’s publication criteria as it currently stands. Therefore, we invite you to submit a revised version of the manuscript that addresses the points raised during the review process.

We would appreciate receiving your revised manuscript by May 31 2020 11:59PM. To enhance the reproducibility of your results, we recommend that if applicable you deposit your laboratory protocols in protocols.io, where a protocol can be assigned its own identifier (DOI) such that it can be cited independently in the future. For instructions see: http://journals.plos.org/plosone/s/submission-guidelines#loc-laboratory-protocols

We look forward to receiving your revised manuscript.

Kind regards,

Chiara Lazzeri

Academic Editor

PLOS ONE

Journal Requirements:

1) Thank you for including your ethics statement:

"The Institutional Review Boards

90 of the four participating hospitals approved the study protocol.".

i) Please amend your current ethics statement to include the full name of the ethics committee/institutional review board(s) that approved your specific study.

ii) Once you have amended this/these statement(s) in the Methods section of the manuscript, please add the same text to the “Ethics Statement” field of the submission form (via “Edit Submission”).

2. In ethics statement in the manuscript and in the online submission form, please provide additional information about the patient records used in your retrospective study. Specifically, please ensure that you have discussed whether all data were fully anonymized before you accessed them and/or whether the IRB or ethics committee waived the requirement for informed consent. If patients provided informed written consent to have data from their medical records used in research, please include this information.

3. Thank you for including your fuinding statement; "This work was supported by the Yuhan Corporation, Republic of Korea (P.C.O.). The funder had no role in study design, data collection and analysis, decision to publish, or preparation of the manuscript."

We note that you received funding from a commercial source:Yuhan Corporation

Reviewers' comments:

Reviewer's Responses to Questions

**Comments to the Author**

1. Is the manuscript technically sound, and do the data support the conclusions?

Reviewer #1: Yes

Reviewer #2: Yes

2. Has the statistical analysis been performed appropriately and rigorously? 

Reviewer #1: Yes

Reviewer #2: Yes

3. Have the authors made all data underlying the findings in their manuscript fully available?

Reviewer #1: Yes

Reviewer #2: Yes

4. Is the manuscript presented in an intelligible fashion and written in standard English?

Reviewer #1: Yes

Reviewer #2: Yes

5. Review Comments to the Author

Reviewer #1: Summary:

In this paper, authors analyzed combined prognostic impact of elevated serum transaminases as a sign of hepatic injury and elevation of ALP (taken at admission) in STEMI patients who underwent primary percutaneous coronary intervention. They retrospectively analyzed data from approximately 1200 patients for 25 months.

Major comments:

1. Elevation of ALP/AST/ALT need not to be caused by hepatic injury during myocardial infarction but by preexisting serious comorbidities affecting also prognosis. Despite authors excluded patients based on the history of serious liver and probably other gastrointestinal diseases I miss data from patients who died during 48 hours after admission and from those not treated by PCI (in particular, excluded by the operator - see Methods, part Primary PCI and in-hospital management).

2. Authors admitted correctly limitations of their approach, but I miss more data, analyses, discussion regarding ALT (rather than AST) as a more specific marker of liver injury, whenelevated. In addition, miss GGT as another important marker and possibly maker (for example, see https://www.ncbi.nlm.nih.gov/pubmed/26463174) for potential liver but also extrahepatal damage.

3. Information about smoking status, treatment by statins, ACE inhibitors is missing and should be included as another part of factors affecting prognosis.

4. I miss the main point of the article - should be recommended different approach to patients with altered liver, … tests (see for example article by Simon TG below)?

Minor comments:

1. In certain aspect the sentence: “To our knowledge, this is the first study to report the usefulness of combined serum transaminases and ALP determined in the emergency room as predictors for MACCE in STEMI” is correct, but authors should mention other studies focused on the impact of liver tests on prognosis in patients with acute coronary syndrome (Simon TG, Corey KE, Cannon CP, Blazing M, Park JG, O'Donoghue ML, Chung RT, Giugliano RP. The nonalcoholic fatty liver disease (NAFLD) fibrosis score, cardiovascular risk stratification and a strategy for secondary prevention with ezetimibe. Int J Cardiol. 2018 Nov 1;270:245-252.) and/or in general population (Golabi P, Stepanova M, Pham HT, Cable R, Rafiq N, Bush H, Gogoll T, Younossi ZM. Non-alcoholic steatofibrosis (NASF) can independently predict mortality in patients with non-alcoholic fatty liver disease (NAFLD). BMJ Open Gastroenterol. 2018;5(1):e000198. Kim D, Kim WR, Kim HJ, Therneau TM. Association between noninvasive fibrosis markers and mortality among adults with nonalcoholic fatty liver disease in the United States. Hepatology. 2013;57(4):1357-65.)

2. To really highlight the additional value of adding liver tests to already established prognostic factors in patients with STEMI (age, EF of LV, …) it is statistically sound to present areas under the receiver operating characteristic curve or other statistical methods for risk/prognostic reclassification.

3. Did authors have also data regarding troponins instead of CK-MB?

4. There are several ALP isoenzymes produced by several body tissues, including the bones, liver, bile - this aspect should be discussed (see also point 1 in Major comments).

5. The term dysglycemia is confusing. Did authors mean hyperglycemia?

6. Regarding patients with STEMI, rather low BMI was observed - in physiological values - is it common in the ethnical group (Asia origin) under study?

7. I miss the point why statistical Model 2 was used or why calcium and phosphate were including - it is rather overstandardization.

Conclusion:

Well described and presented study focused on recently very popular topic of liver involvement as prognostic factor, cardiovascular disease, in this case in patients with ST-elevation myocardial infarction. Nevertheless, this study should be complemented by important data/analyses

Reviewer #2: In this manuscript Kang et al investigated the prognostic role of the combination of serum transaminase and alkaline phosphatase (ALP) in 1176 STEMI patient treated with primary PCI. The authors showed that patients with hypoxic liver injury (high serum transaminase) and high ALP had major MACCE at median follow-up and these two parameters were independent prognostic factors. The research topic was original because in literature there are some evidence about prognostic role of serum transaminase in STEMI patients and poor evidence of prognostic role of ALP in STEMI patients but never two parameters were combined together. The study population, from INTERSTELLAR Registry, was large and homogeneous and final results were significant. Nevertheless some limitations should be highlighted such as the retrospective and observational study design and cut-off value of ALP (median value of ALP).

Some considerations should be made:

- The authors chose ALP as prognostic parameter for this study. Why did they choose ALP instead of other parameters for acute liver injury such as bilirubin?

- In this study serum transaminase were measured only at admission, conversely in literature other authors showed the importance of serum transaminase trend during acute hospital length for STEMI patients( Lazzeri C, Valente S, Boddi M, Mecarocci V, Chiostri M, Gensini GF. Clinical and prognostic significance of increased liver enzymes in ST-elevation myocardial infarction. Int J Cardiol. 2014 Dec 15;177(2):543-4.) The authors should add this consideration in the paragraph “limitation”.

- In this manuscript serum transaminase and ALP were measured at the emergency department but STEMI patients nowadays should bypass the emergency department and should be admitted directly at catheterization laboratory, as suggested by international guidelines. The authors should clarify internal protocol for STEMI patients.

- The follow-up data were limited at 36 and 48 months

- Lines 163-164: the numerical sum is 99.9%, the authors should optimize numerical aproximations

6. PLOS authors have the option to publish the peer review history of their article (what does this mean?). If published, this will include your full peer review and any attached files.

Reviewer #1: No

Reviewer #2: No

---

## [Author Response · Author response to Decision Letter 0]

29 Apr 2020

We would like to thank you for the kind and thorough comments on our manuscript entitled, "Prognostic impact of the combination of serum transaminase and alkaline phosphatase determined in the emergency room in patients with ST-segment elevation myocardial infarction undergoing primary percutaneous coronary intervention".

We have revised our manuscript, wherever possible, based on the reviewer’s comments and we feel this has strengthened our paper considerably. The changes and responses to specific comments are detailed below.

Reviewer #1: Summary:

In this paper, authors analyzed combined prognostic impact of elevated serum transaminases as a sign of hepatic injury and elevation of ALP (taken at admission) in STEMI patients who underwent primary percutaneous coronary intervention. They retrospectively analyzed data from approximately 1200 patients for 25 months.

Major comments:

1. Elevation of ALP/AST/ALT need not to be caused by hepatic injury during myocardial infarction but by preexisting serious comorbidities affecting also prognosis. Despite authors excluded patients based on the history of serious liver and probably other gastrointestinal diseases I miss data from patients who died during 48 hours after admission and from those not treated by PCI (in particular, excluded by the operator - see Methods, part Primary PCI and in-hospital management).

We totally agree to your comments. Elevation of serum transaminases or ALP could be related with pre-existing hepatobiliary disease (such as chronic viral hepatitis, liver cirrhosis, or biliary stone and infection), bone disease, or CKD. So, we completely excluded these patients (n=275). This registry included only patients with STEMI who underwent primary PCI. Patients who were treated by thrombolytic therapy, or died before primary PCI, were excluded. After primary PCI, in-hospital death occurred in 37 patients (3.1%) and median duration of hospitalization for these patients was 4.0 days (interquartile range, 1.0-16.5 days). We added this data at the Results. Patients who died during 48 hours after admission were 12. Unfortunately, we could not get data from patients not treated by primary PCI, because this registry did not include these patients.

2. Authors admitted correctly limitations of their approach, but I miss more data, analyses, discussion regarding ALT (rather than AST) as a more specific marker of liver injury, when elevated. In addition, miss GGT as another important marker and possibly maker (for example, see https://www.ncbi.nlm.nih.gov/pubmed/26463174) for potential liver but also extrahepatal damage.

Serum ALT is found primarily in the liver and a more specific marker of liver injury. Serum AST is present not only in the liver but also in other organs, including the heart, skeletal muscle, kidney, and brain. Therefore, whether AST were elevated solely due to hypoxic liver damage is not certain. In this study, elevation of serum ALT or AST > twice the upper limit of normal was in 71 patients (6.0%) or 205 patients (17.4%), respectively. We analyzed clinical outcomes according to the levels of serum ALT (except serum AST) and ALP and added this data as the S2 Table showing a significant trend to higher incidence of in-hospital mortality and MACCE. However, the number of patients who had serum ALT level > twice the upper limit of normal was small and we could calculate only p value for trend. If we use only serum ALT level as the marker of liver injury, specificity for diagnosis of HLI will increase and sensitivity will decrease. The definition of HLI is not fully validated. In this study, we used both serum AST and ALT as the cut-off value of HLI. We added these comments at the Results and Discussion (Limitations) sections.

We totally agree that serum GGT may be another important marker for liver injury. Unfortunately, this registry does not include serum GGT level, so we cannot analyze the prognostic impact of serum GGT in this study. 

3. Information about smoking status, treatment by statins, ACE inhibitors is missing and should be included as another part of factors affecting prognosis.

We added data about smoking status and discharge medication at the table 1 and the S1 Table, respectively. There were no significant differences of discharge medications including beta-blocker, renin-angiogensin system blocker and statin among the 4 groups (S1 Table).

4. I miss the main point of the article - should be recommended different approach to patients with altered liver, … tests (see for example article by Simon TG below)?

We suggested that the combined information on HLI and serum ALP level might serve as an early prognosticator in STEMI. Therefore, high risk patients with HLI and high ALP level are most likely to derive benefit from early aggressive treatment for cardiac pump failure and cardiovascular risk factors, such as early initiation of renin-angiotension system blocker or high-intensity statin. However, a potential method to improve the prognosis of patients based on the HLI status and serum ALP levels is beyond the scope of this study. We added this comment at the Discussion.

Minor comments:

1. In certain aspect the sentence: “To our knowledge, this is the first study to report the usefulness of combined serum transaminases and ALP determined in the emergency room as predictors for MACCE in STEMI” is correct, but authors should mention other studies focused on the impact of liver tests on prognosis in patients with acute coronary syndrome (Simon TG, Corey KE, Cannon CP, Blazing M, Park JG, O'Donoghue ML, Chung RT, Giugliano RP. The nonalcoholic fatty liver disease (NAFLD) fibrosis score, cardiovascular risk stratification and a strategy for secondary prevention with ezetimibe. Int J Cardiol. 2018 Nov 1;270:245-252.) and/or in general population (Golabi P, Stepanova M, Pham HT, Cable R, Rafiq N, Bush H, Gogoll T, Younossi ZM. Non-alcoholic steatofibrosis (NASF) can independently predict mortality in patients with non-alcoholic fatty liver disease (NAFLD). BMJ Open Gastroenterol. 2018;5(1):e000198. Kim D, Kim WR, Kim HJ, Therneau TM. Association between noninvasive fibrosis markers and mortality among adults with nonalcoholic fatty liver disease in the United States. Hepatology. 2013;57(4):1357-65.)

Thank you for introducing the brilliant studies. We mentioned these studies at the Introduction as follows:

Recent studies suggest that nonalcoholic fatty liver disease fibrosis score including serum transaminases is a significant predictor of mortality in general population [3, 4]. This score is associated with increased risk of recurrent cardiovascular events in post-acute coronary syndrome patients [5].

2. To really highlight the additional value of adding liver tests to already established prognostic factors in patients with STEMI (age, EF of LV, …) it is statistically sound to present areas under the receiver operating characteristic curve or other statistical methods for risk/prognostic reclassification.

We analyzed C-statistics in the multivariate Model 1 for predicting MACCE. Then, after adding HLI and ALP status into the Model 1, the difference of c-statistics was calculated. This information was added at the Results and S1 figure as follows:

C-statistics for predicting MACCE was 0.656 (95% CI, 0.604-0.708) in the multivariate Model 1 excepting HLI status and high ALP. The addition of HLI status and high ALP to the Model 1 significantly improved the c-statistics to 0.685 (95% CI, 0.635-0.736; p for difference = 0.044) (S1 Fig).

3. Did authors have also data regarding troponins instead of CK-MB?

This study analyzed data from 4-regional hospital based registry. Blood tests were not measured using a central laboratory. So, we could not use data of troponins because the method of measuring troponin in each hospital is different (troponin I or T and high-sensitivity or not). 

4. There are several ALP isoenzymes produced by several body tissues, including the bones, liver, bile - this aspect should be discussed (see also point 1 in Major comments).

We added this comment at the Discussion (Prognostic implication of serum ALP level) as follows:

ALP is actually enzyme found primarily liver and bone, and to lesser extents in intestine, placenta, and kidneys. Accordingly, serum ALP is used as a marker of hepatic or bony disease in clinical practice. ~~~

~~~ However, bone-specific ALP levels were not available in this study, and this mechanism remains theoretical.

5. The term dysglycemia is confusing. Did authors mean hyperglycemia?

In the study of Jang HJ et al. (Am J Cardiol. 2017;119:1179-85.), dysglycemia was defined as either hypoglycemia (serum glucose <90 mg/dl) or hyperglycemia (serum glucose >250 mg/dl). So, we changed dysglycemia to dysglycemia (hyper or hypo-glycemia).

6. Regarding patients with STEMI, rather low BMI was observed - in physiological values - is it common in the ethnical group (Asia origin) under study?

Yes. Asian patients generally have lower BMI than western patients. In another study, mean BMI in Korean patients with STEMI (n=6,246) was 24.1 ± 3.4 kg/m2 (Kim JH et al, Circ J. 2016;80:1427-36).

7. I miss the point why statistical Model 2 was used or why calcium and phosphate were including - it is rather overstandardization.

We agree to your comments. Serum ALP level is elevated in bone disease, such as renal osteodystrophy, affecting calcium and phosphorus metabolism. Tonelli M et al. reported that the excess risk of death was particularly high among people with higher levels of both ALP and serum phosphate (Tonelli M et al. Circulation. 2009;120:1784-92.). So, we put serum calcium and phosphate into the statistical Model 2. However, there were no significant differences of serum calcium and phosphate levels among 4 groups. We removed these variables in the Model 2 to avoid overstandardization. 

Conclusion:

Well described and presented study focused on recently very popular topic of liver involvement as prognostic factor, cardiovascular disease, in this case in patients with ST-elevation myocardial infarction. Nevertheless, this study should be complemented by important data/analyses

Reviewer #2: In this manuscript Kang et al investigated the prognostic role of the combination of serum transaminase and alkaline phosphatase (ALP) in 1176 STEMI patient treated with primary PCI. The authors showed that patients with hypoxic liver injury (high serum transaminase) and high ALP had major MACCE at median follow-up and these two parameters were independent prognostic factors. The research topic was original because in literature there are some evidence about prognostic role of serum transaminase in STEMI patients and poor evidence of prognostic role of ALP in STEMI patients but never two parameters were combined together. The study population, from INTERSTELLAR Registry, was large and homogeneous and final results were significant. Nevertheless some limitations should be highlighted such as the retrospective and observational study design and cut-off value of ALP (median value of ALP).

Some considerations should be made:

- The authors chose ALP as prognostic parameter for this study. Why did they choose ALP instead of other parameters for acute liver injury such as bilirubin?

This is a very important question. We analyzed liver function parameters including albumin, total bilirubin, AST, ALT and ALP between patients with MACCE and without MACCE. Serum AST, ALT and ALP were significantly higher in patients with MACCE, compared to those in patients without MACCE. There was no significant difference of total bilirubin level. Thus, we chose serum transaminase and ALP among liver function parameters. Unfortunately, serum gamma-GT levels were not available in this study. 

- In this study serum transaminase were measured only at admission, conversely in literature other authors showed the importance of serum transaminase trend during acute hospital length for STEMI patients( Lazzeri C, Valente S, Boddi M, Mecarocci V, Chiostri M, Gensini GF. Clinical and prognostic significance of increased liver enzymes in ST-elevation myocardial infarction. Int J Cardiol. 2014 Dec 15;177(2):543-4.) The authors should add this consideration in the paragraph “limitation”.

We added this comment in the paragraph “limitation” as follows:

Fifth, we did not obtain serial data of serum transaminases and ALP levels during the hospitalization. Peak values of serum transaminases were significantly more related with in-hospital mortality than admission values in STEMI patients [23]. It was impossible to determine its prognostic implication.

- In this manuscript serum transaminase and ALP were measured at the emergency department but STEMI patients nowadays should bypass the emergency department and should be admitted directly at catheterization laboratory, as suggested by international guidelines. The authors should clarify internal protocol for STEMI patients.

In Korea, most patients suspected of STEMI are admitted to the ER first. Then, if ST-segment elevation is noted at the EKG, catheterization laboratory will be activated sequentially. Nowadays, in some hospitals, patients diagnosed with STEMI before arriving at the hospital can admit directly to the CCU or catheterization laboratory. However, in this study (between 2007 and 2014), all patients admitted to the ER first. We added this protocol for STEMI patients at the Method as follows:

All patients visited the emergency room first without being admitted directly to the catheterization laboratory.

- The follow-up data were limited at 36 and 48 months

We totally agree to your comment.

- Lines 163-164: the numerical sum is 99.9%, the authors should optimize numerical aproximations

There was an error in rounding the percentage. We corrected the percentage as follows:

116 patients (9.8% � 9.9%) were in the HLI (+) & high ALP group.

---

## [Editor Report · Decision Letter 1]

4 May 2020

Prognostic impact of the combination of serum transaminase and alkaline phosphatase determined in the emergency room in patients with ST-segment elevation myocardial infarction undergoing primary percutaneous coronary intervention

PONE-D-19-32757R1

Dear Dr. Kang,

We are pleased to inform you that your manuscript has been judged scientifically suitable for publication and will be formally accepted for publication once it complies with all outstanding technical requirements.

With kind regards,

Chiara Lazzeri

Academic Editor

PLOS ONE
---

## [Editor Report · Acceptance letter]

8 May 2020

PONE-D-19-32757R1 

Prognostic impact of the combination of serum transaminase and alkaline phosphatase determined in the emergency room in patients with ST-segment elevation myocardial infarction undergoing primary percutaneous coronary intervention 

Dear Dr. Kang:

I am pleased to inform you that your manuscript has been deemed suitable for publication in PLOS ONE. Congratulations! Your manuscript is now with our production department. 

With kind regards,

on behalf of

Dr. Chiara Lazzeri 

Academic Editor

PLOS ONE